# Third Generation Buchwald Precatalysts with XPhos and RuPhos: Multigram Scale Synthesis, Solvent-Dependent Isomerization of XPhos Pd G3 and Quality Control by ^1^H- and ^31^P-NMR Spectroscopy

**DOI:** 10.3390/molecules26123507

**Published:** 2021-06-09

**Authors:** Svitlana O. Sotnik, Artem M. Mishchenko, Eduard B. Rusanov, Andriy V. Kozytskiy, Konstantin S. Gavrilenko, Sergey V. Ryabukhin, Dmitriy M. Volochnyuk, Sergey V. Kolotilov

**Affiliations:** 1Enamine Ltd., 78 Chervonotkatska Street, 02660 Kyiv, Ukraine; sotniksvitlana@ukr.net (S.O.S.); a.m.mishchenko@ukr.net (A.M.M.); kozytskiy@gmail.com (A.V.K.); kgavrio@gmail.com (K.S.G.); s.v.ryabukhin@gmail.com (S.V.R.); d.volochnyuk@gmail.com (D.M.V.); 2L.V. Pisarzhevskii Institute of Physical Chemistry, National Academy of Sciences of Ukraine, Nauky Avenue 31, 03028 Kiev, Ukraine; 3Institute of High Technologies, National Taras Shevchenko University of Kyiv, 60 Volodymyrska Street, 01033 Kyiv, Ukraine; 4V.I. Vernadsky Institute of General and Inorganic Chemistry, National Academy of Sciences of Ukraine, Palladina Avenue 32/34, 03142 Kiev, Ukraine; 5Institute of Organic Chemistry, National Academy of Sciences of Ukraine, Murmanska Street 5, 03028 Kiev, Ukraine; XRAY@ioch.kiev.ua

**Keywords:** palladium, Buchwald precatalysts, NMR spectroscopy, quality control

## Abstract

The third generation Buchwald precatalysts Pd(ABP)(Phos)(OMs) (also known as Phos Pd G3)) with XPhos and RuPhos were prepared in multigram scale by a modified procedure (ABP = fragment of C-deprotonated 2-aminobiphenyl, XPhos = 2-dicyclohexylphosphino-2′,4′,6′-triisopropylbiphenyl, RuPhos = 2-dicyclohexylphosphino-2′,6′-diisopropoxybiphenyl, OMs^−^ = CH_3_SO_3_^−^). The ^1^H- and ^31^P-NMR spectra of the title complexes and some impurities, measured by various 1D and 2D techniques, were analyzed in detail. The solvent-dependent isomerization of Pd(ABP)(XPhos)(OMs) was studied by NMR, and the X-ray structures of two isomers were determined. The impurities in precatalysts, such as Pd(ABP)(HABP)(OMs) (HABP—neutral 2-aminobiphenyl coordinated to Pd^2+^ in *N*-monodentate mode) and PdCl_2_(XPhos)_2_, were identified and characterized by single crystal X-ray diffraction. A simple method for the quick quality control (QC) of the precatalysts, suitable for routine use, was proposed. The method was based on the assessment of the impurity content on the basis of the ^1^H-NMR spectra analysis.

## 1. Introduction

Palladium complexes with phosphine ligands are widely used catalysts in fine organic synthesis, particularly for C–C [1,2,3] and C–heteroatom [4,5] cross-coupling, homocoupling [6,7] and metathesis [8,9] reactions. During the last two decades, significant advances were achieved in the development of precatalysts of this type, and at present, several generations of the precatalysts are distinguished [1,2,4]. Among them, Buchwald precatalysts based on dialkylbiaryl phosphines of the third generation (G3)—so-called “Phos Pd G3 precatalysts” of composition Pd(ABP)(Phos)(OMs) (where ABP is the residue of 2-aminobiphenyl after one H atom elimination, Phos is specially designed phosphine, OMs^−^ = CH_3_SO_3_^−^, Figure 1)—are of special interest due to their high catalytic activity compared to complexes of more “simple” structures, such as classical Pd(PPh_3_)_4_ [1,4,10]. The advantages of PdG3(Phos) systems include high activity along with stability in solutions and reasonable tolerance to air and moisture [11,12,13]. In the reaction mixture, Pd(ABP)(Phos)(OMs) complexes undergo reductive transformation producing active PhosPd^(0)^ species, which directly participate in the catalytic cycle [3,4] (for this reason, the complexes Pd(ABP)(Phos)(OMs) are usually referred to as “the precatalysts”, which form “the catalysts” in situ).

Catalyst quality plays an important role in the success of the catalytic transformations. From our own experience, there are several reasons to control the quality of the Pd(ABP)(Phos)(OMs) precatalyst when it is purchased from a commercial source or synthesized in laboratory for in-house scientific research purposes [14]. First, the sample may contain unreacted starting materials, such as [Pd(ABP)(OMs)]_2_, or other nonidentified coordination compounds, such as phosphine oxide, that, in the best case, play a role of inert “ballast” but can also induce unwanted side reactions. Second, the sample may contain residues of solvents used in its synthesis, especially solvents such as CH_2_Cl_2_, which appear to introduce unwanted reactive impurity for some processes. Third, partial decomposition of the complex may take place if the storage conditions are not complied with, leading to reduced catalytic performance of the sample. In all these cases, different samples may have significantly different catalytic activity, which cannot be foreseen. It is important to note that common methods for the purification of coordination compounds are not efficient in this case, because the complexes of the Pd(ABP)(Phos)(OMs) family are sensitive to air and heating. For example, our multiple attempts to purify such compounds by recrystallization usually gave the samples a similar purity (presumably due to formation of new impurities during the process); drying of the solids in vacuo to remove residues of the solvent almost never gave the desired result, but sometimes led to the partial decomposition of the sample. For these reasons, in-house production of the precatalysts for research purposes can be a more simple alternative compared to the analysis of commercially available samples or the elimination of the residues of specific solvents which are not desired for a certain application (instead, other solvent can be used for synthesis, the residues of which are not harmful for further stages of the catalyst use).

Therefore, the development of a simple method for estimating the purity of the precatalysts, which can be used for their rapid everyday quality control (QC), is an important task. The problem of QC in this case cannot be effectively solved using many physical and physicochemical methods, because they are not sensitive enough to detect even 12% of impurities (elemental analysis, infrared or electronic spectroscopy) or do not provide reliable information on the impurities (single crystal XRD analysis of an arbitrarily chosen crystal; powder X-ray diffraction of bulk sample, since the impurity may be noncrystalline or chromatography with mass spectrometry, since the impurity may decompose on the column, giving fragments with high volatility that merge with the solvent or do not give distinct patterns in the MS). Thus, the determination of the impurities by many methods can be labour- and time-consuming and require complex equipment, while not giving reliable results.

In order to develop a simple and reliable method for the QC, we examined the NMR spectra of the Phos Pd G3 precatalysts. A similar approach was previously proposed to analyse the purity of Pd(II) acetate [15,16,17] and the complex of Pd(0) with dibenzylideneacetone [18], and in our experience, these methods appeared to be very efficient for the regular QC of these compounds.

In the present study, the possibility of estimating the purity of the G3 precatalysts Pd(ABP)(Phos)(OMs) with 2-dicyclohexylphosphino-2′,4′,6′-triisopropylbiphenyl (XPhos) and 2-dicyclohexylphosphino-2′,6′-diisopropoxybiphenyl (RuPhos) by ^1^H- and ^31^P-NMR was demonstrated, and equations to determine the percentage of impurities in these compounds on the basis of their ^1^H-NMR spectra were derived. For this aim, the majority of signals in NMR spectra were assigned to certain nuclei using various techniques of 1D and 2D NMR spectroscopy. The reversible isomerization of Pd(ABP)(Phos)(OMs) upon solvent change was studied. Taking in mind the high importance and wide use of the Buchwald precatalysts in modern organic chemistry, we also isolated and identified several impurities in the precatalysts. The knowledge of the structure of such impurities can be helpful for the analysis of possible side reactions in fine organic synthesis.

## 2. Results and Discussion

### 2.1. Synthesis and X-ray Structures of the Complexes

The third generation Buchwald precatalysts Pd(ABP)(XPhos)(OMs) are commonly prepared in three steps [11]. At the first step, 2-ammoniumbiphenyl mesylate is obtained from 2-aminobiphenyl and methanesulfonic acid [11]. Then, dimeric palladacycle [Pd(ABP)(Oms)]_2_ (**1**) is synthesized from palladium acetate and 2-ammoniumbiphenyl mesylate.

From the solution remaining after crystallization of **1**, we obtained complex **2**, isolated as two solvates **2a** and **2b** (i.e., Pd(ABP)(HABP)(OMs)·0.12C_6_H_5_CH_3_·0.33MTBE and [Pd(ABP)(HABP)(CH_3_CN)](OMs)·H_2_O, respectively (see Experimental Section for details). Compound **2** could form upon the reaction of palladium (II) acetate with HABP in a 1:2 molar ratio or in the reaction of **1** with the excess HABP. In the second case, HABP, acting just as a monodentate amine, caused the cleavage of **1**, forming two mononuclear complexes. The possibility of such cleavage is clearly illustrated by the formation of Pd(ABP)(Phos)(OMs) complexes upon the reaction of **1** with phosphines, such as XPhos and RuPhos, because this reaction involves a similar cleavage. The formation of similar complexes Pd(ABP)(py)(OMs), where py is pyridine, was also reported [11].

The structure of **2**, crystallized from acetonitrile, compound **2b**, was determined by single crystal X-ray diffraction (Figure 2). In **2b**, the Pd^2+^ ion is located in a distorted square planar donor set CN_3_. The deprotonated 2-aminobiphenyl (ABP) is formally monoanionic and coordinated in a C,N-bidentate mode, whereas the HABP molecule acts as a neutral N-monodentate ligand. The acetonitrile nitrogen atom occupies the fourth site in the coordination sphere, completing the coordination environment of Pd^2+^ to square the planar CN_3_ donor set. The resulting positive charge of the [Pd(ABP)(HABP)(CH_3_CN)]^+^ cation is counterbalanced by the outersphere mesylate anion. The bond length and angles in the PdCN_3_ chromophore are typical for palladium(II) complexes (Table 1) [11,19]. Quite unexpected, the coordinated ABP ligands adopt such configuration that almost all atoms of the ligands are located on one side from the PdCN_3_ plane, while the axial position over the Pd^2+^ ion is occupied by the C atom of the CH_3_ group of acetonitrile (d(Pd-C) is 3.8 Å). Such localization of the CH_3_ group is probably caused by dispersion forces.

At the next step of the precatalyst synthesis, 1 mole of dimer **1** is treated with 2 moles of the corresponding phosphine (such as XPhos or RuPhos) [11]. In the reported procedures, THF or DCM were used as the media for the reactions, depending on the phosphine ligand. THF was recommended for XPhos Pd G3 and RuPhos Pd G3 [11]. However, we have found that the solid products prepared by standard procedure retained THF in different and irreproducible amounts, and the solvent could not be removed even after heating the sample for 7 h at 80 °C in an oil pump vacuum (0.08 torr).

One way to remove THF from the crystal lattice of a sample is to replace it with a more volatile solvent, such as dichloromethane (DCM). For example, when the sample of the composition Pd(ABP)(XPhos)(OMs)·xTHF was dissolved in DCM and the resulting solution was evaporated in vacuum, the product Pd(ABP)(XPhos)(OMs)·0.15THF·1.4DCM was obtained. Repeating the procedure led to the almost complete removal of THF and the formation of Pd(ABP)(XPhos)(OMs)·0.01THF·0.6DCM. Heating it in vacuum at 70 °C for 1 h caused a 3-fold decrease of the DCM content, while the vacuum drying at 85 °C yielded Pd(ABP)(XPhos)(OMs)·0.01THF·0.1DCM. Thus, removal of the residual solvent from the complex bearing branched dialkylbiaryl phosphines is a nontrivial task.

The strong retention of solvent in the crystal lattice of XPhos Pd G3 and RuPhos Pd G3 complexes may be caused by the formation of H-bonds as well as by solvent capture in the pockets of the crystal lattice. Notably, there are signals of THF in the published spectra of XPhos Pd G3 and RuPhos Pd G3 measured in CD_3_OD and CDCl_3_, respectively [11]. It is also important that the complexes **3** and **5**, reported in this paper, had the same catalytic activity in C-C and C-N coupling reactions [14] as the compounds synthesized by the known procedures [11]. In our opinion, in large-scale synthesis, it is more appropriate to control the solvent composition of samples rather than to spend efforts on their desolvation.

The X-ray structures of two compounds, prepared by crystallization of PdG3(XPhos)·0.18THF from pure ethyl acetate or from acetone containing ca. 0.5% of THF (added to increase solubility of the complex), were analyzed. The resulting compounds (**3a** and **3b**) appeared to differ not only by solvate composition, but by the mode of XPhos coordination to Pd^2+^.

The complex **3a** (Figure 3), obtained from ethyl acetate, is isostructural to the previously reported solvate of XPhos Pd G3 (for **3a** a = 12.3572(11) Å, b = 13.6870(13) Å, c = 14.6964(13) Å, α = 101.015(6)°, β = 94.793(6)°, γ = 99.427(5)°, while for the reported analogue [11] a = 12.3071(10) Å, b = 13.5653(11) Å, c = 14.7831(12) Å, α = 102.4550(10)°, β = 94.4810(10)°, γ = 98.6540(10)°, both P-1 space group). The coordination environment of the Pd^2+^ ion in **3a** is formed by the C,N atoms of the ABP^−^ anion, the O atom of the coordinated mesylate and the P atom of XPhos; the latter is coordinated in monodentate mode.

Though the solvent used in [11] for single crystals growth was not indicated, the crystal lattice of the previously reported XPhos Pd G3 contains diethyl ether. In the crystal lattice of **3a**, ethyl acetate was localized, but no H-bonds were found between the molecules of this solvent and the polar groups of the complex compound (Appendix A, Appendix A). These results cannot be directly transferred to the description of the cases of THF or DCM (the solvents captured in the synthesis of **3**, see Experimental Section); however, from analysis of the crystal structure of **3a** and previously reported XPhos Pd G3 [11], we can note that there are large voids in the crystal packing of **3a** suitable for the capture of solvent, and the crystal packing is not sensitive to the nature of the solvent (ethyl acetate or diethyl ether). These voids do not form infinite channels, allowing for the free movement of the captured solvent molecules; this observation can explain the strong retention of the captured molecules and the difficulties of sample desolvation.

The structure of complex **3b** (Figure 4), obtained from acetone containing a minor quantity of THF, is very different from that of **3a**. Despite the low quality of single crystals (twinning and severe disorder of both mesylate and the captured solvent), the main features of the structure of **3b** were reliably determined. The OMes^−^ anion is not coordinated to the Pd^2+^ ion; however, XPhos is bound to Pd^2+^ through the P atom and C=C bond of the 2,4,6-triisopropylbenzene ring. Thus, the coordination number of the Pd^2+^ ion is four; three positions are filled by the C,N atoms of ABP^−^ and the P atom of XPhos, and the fourth position is filled due to π-bonding of the same XPhos though a C=C bond. However, the plane PdNPC_2_ includes the C19 atom, while the C20 atom lies out of the plane (d(Pd1-C20) is 0.123 Å longer compared to d(Pd1-C19), see Table 2), and the C19-Pd1-C12 angle is 174.7(3)° (which is closer to 180° compared to 150.1(3)° for C12-Pd1-C20 angle). This mode of Xphos coordination in **3b** led to a slight decrease of Pd-P and Pd-N bonds compared to **3a** (Table 2). Selected structural parameters of **3a** and **3b** are listed in Table 2. Similar structures of Phos Pd G3 complexes with π-bond between the Pd^2+^ ion and C=C bonds of the aromatic rings were reported for several complexes in [11]. We suppose that other isomers of the same compounds can form upon proper solvent selection.

An attempt to determine the structure of impurity **4** formed in synthesis of **3** by X-ray single crystal structure analysis was performed. Single crystals of compound **4**, sufficient for X-ray quality, could be obtained by crystallization from chloroform. The complex which formed contained chloride ions, coordinated to the Pd^2+^ ion; such chloride presumably formed due to the decomposition of CHCl_3_ (catalytic decomposition at presence of palladium(II) complexes cannot be excluded). The molecule is centrosymmetric (Figure 5), and the Pd^2+^ ion is located in the inversion center in square planar PdCl_2_P_2_ donor set. The bond lengths and angles have values typical for palladium(II) complexes (Table 3).

The molecular structure of **5** (Figure 6) was similar to **3a**. RuPhos is coordinated to the P atom and C=C bond of the RuPhos ligand, similar to **3b**. The mesylate anion is not coordinated to the Pd^2+^ ion. Thus, the Pd^2+^ ion is located in the square planar C_2_PN donor set. Bond lengths and angles have values typical for palladium(II) complexes (Table 4). Among C atoms of the phenyl ring, separation of the Pd^2+^ ion from C19 is much lower compared to distances between the Pd^2+^ ion and C20 or C24 atoms (d(Pd-C19) is 2.421(4) Å, while d(Pd-C20) and d(Pd-C24) are 2.743(4) and 2.948(3) Å, respectively).

Discussion of the role of solvent in the formation of certain isomers of the Phos Pd G3 complex will be presented below.

### 2.2. ^1^H-NMR Spectra

The quality of a large number of samples can be conveniently controlled using ^1^H-NMR spectroscopy by comparison of the integrated intensities of signals attributed to impurities and the main component [18]. However, in order to apply this technique to the G3 Buchwald precatalysts, their spectra should be studied in details. The ^1^H-NMR spectra were concisely reported without discussion in the seminal paper [11], and to the best of our knowledge, no other research dealing with such spectra was published. To fill this gap, as well as to obtain reliable information for the identification of various species in solutions, the NMR spectra of Pd G3 precatalysts with two phosphines (XPhos and RuPhos) were analyzed in detail.

In the preliminary experiments, it was shown that informative spectra could be obtained at routine conditions if the samples of the studied complexes were dissolved in a polar solvent (such as CD_3_OD or DMSO-*d*_6_) at a concentration not less than 30 mmol/L. Since XPhos Pd G3 and RuPhos Pd G3 are stable in DMSO-*d*_6_ solution, assignment of proton signals and applying of ^1^H-NMR spectroscopy for QC of these precatalysts was performed using NMR data obtained by analysis of the appropriate spectra measured in DMSO-*d*_6_ solutions.

The signals of 10 protons (eight aromatic and two H atoms of amino group, vide infra) in the ^1^H-NMR spectrum of **1** are arranged in several groups of signals, evidencing that two ABP moieties in **1** are equivalent or very similar (Figure 7). In contrast, the ^1^H-NMR spectrum of **2a** (Figure 7) exhibits signals corresponding to 17 protons (nine of them can be assigned to neutral ligand and eight to monoanionic one) in the region of H atoms of the aromatic system. It can be noted that the structures of **1** and **2** are retained in solution in DMSO-*d*_6_, since all signals correspond to the expected ones, and there are no signs of dissociation. There are also no signs of equilibrium between **1** and **2** in solution—there are no signals of **1** in the spectrum of **2**, and vice versa. Dissociation of mesylate associated with the formation of additional bonds of the Pd^2+^ ion with bulky phosphine ligand in solution was reported for tBuXPhos Pd G3 and BrettPhos Pd G3 [11].

In order to see how isomeric species formation depends on the solvent nature, ^31^P-NMR spectra of the XPhos Pd G3 were measured in the mixture of solvents, the composition of which varied from pure acetone to pure ethyl acetate (Figure 8). There was one sharp signal of ^31^P in acetone at δ = 36.1 ppm, and a weak wide signal of the same nuclei at δ = 63.5 ppm. In contrast, in ethyl acetate, the upfield weak signal at 35 ppm was detected with intensity slightly above the noise level, whereas the downfield signal at δ = 62.8 ppm was the most intense, and a weak wide signal at ~59 ppm also appeared. An increase of the content of ethyl acetate in acetone led to a redistribution of the intensity of these signals: the upfield peak gradually became less intense while the downfield peak became more intense (Figure 8). According to X-ray diffraction data there was a Pd—C=C coordination bond in XPhos Pd G3 crystalized from acetone (Figure 4), while the isomer crystallized from ethyl acetate did not contain such a bond (Figure 3). Taking into account that there was a significant difference between the NMR spectra of this complex in acetone and ethyl acetate, and there were only single sharp signals in these solvents (indicating the existence of one form in solution), we assign the spectrum of XPhos Pd G3 in acetone to the form containing the Pd—C=C coordination bond, while the spectrum of this complex in ethyl acetate can be assigned to the form which does not contain such Pd—C=C bond.

We can conclude that two forms of XPhos Pd G3 in solution can be transformed, one into the other, depending on the solvent. Such transformation can be associated with a change in the relative Pd-X bond energies (where X = P or O from OMes^−^ or C=C bond from XPhos) due to different media polarity or solvation effects. Discussion of the ^31^P-NMR signal assignment will be provided in the next section.

The sharp signal at 42.71 ppm was observed in the ^31^P-NMR spectrum of RuPhos Pd G3 in acetone solution, while two broad signals at 39.78 ppm and 49.01 ppm were observed in the ^31^P-NMR spectrum of RuPhos Pd G3 in ethyl acetate solution (Figure 9). Similar to the abovementioned assignment of the XPhos Pg G3 spectra, the signals of RuPhos Pd G3 in acetone can be assigned to the form which contains the Pd—C=C bond (Figure 6), while the spectrum of this complex in ethyl acetate probably contains the signals of other isomers, such as conformation isomers of the RuPhos Pd G3 containing a ligand bound through the Pd-P bond or the species containing a coordinated solvent or counterion. No signal of free RuPhos was detected in the solution (expected at −9.3 ppm, Appendix A), indicating that dissociation of the Pd-P bond did not occur.

There was only one sharp signal at 36.84 ppm and 42.01 ppm in the ^31^P spectra of XPhos Pd G3 and RuPhos Pd G3, respectively, in DMSO-*d*_6_ solution. The proximity of the chemical shift values of the ^31^P signals of these complexes measured in acetone and DMSO-*d*_6_ solutions indicates that in these solvents, the complexes have the same structure with the Pd—C=C coordination bond.

The spectra of XPhos Pd G3 and RuPhos Pd G3 in CDCl_3_ were reported [11]. There were two sharp signals in the spectrum of XPhos Pd G3 at 65.38 and 36.10 ppm, which are very close to the signals observed for this complex in ethyl acetate and acetone, respectively. The presence of two signals can be a sign of the existence of two isomeric forms in solution in CDCl_3_. In the case of RuPhos Pd G3, there was only one signal at 41.57 ppm [11], which is close to the signal found in solution in acetone in our study. This signal can be assigned to the form containing the Pd—C=C bond.

The molecule of Xphos Pd G3 contains 62 H atoms, while RuPhos Pd G3 contains 56 H atoms, giving signals in the −0.1–8.0 ppm range in ^1^H-NMR spectra. In contrast to the spectra of dimer [11] and the corresponding dialkylbiaryl phosphine [20,21], the spectra of the complexes are more complicated. Resonances from different groups superimpose on one another, forming multiplets with a complex structure, which makes assignment more difficult. Several conclusions regarding the structure of RuPhos Pd G3 in solution could be made using ^1^H-^13^C HMBC, ^1^H-^31^P-HMBC and 1D ROESY correlation spectra. These techniques were not so efficient for assigning signals in the ^1^H-NMR spectra of XPhos Pd G3 due to a large number of the strong overlapping signals. Even when we used the PSYCHE experiment for XPhos Pd G3 in tandem with other correlation methods (Appendix A, Appendix A), we did not manage to assign the signals of such complexes in the aliphatic region, since each proton in the cyclohexane nuclei was magnetically nonequivalent to all others. In the case of RuPhos Pd G3, we could assign all observed signals of protons except protons in the cyclohexane fragments (see Figure 10 for H numeration scheme and Appendix A for correlation spectra used for signal assignment). In the case of XPhos Pd G3, the aromatic protons of the phosphine ligand, signal of one proton of amino group and protons of isopropyl groups could be assigned, while the positions of the protons of cyclohexane nuclei and aminobiphenyl could not be specified (Figure 10).

The 1D ROESY experiment was carried out in order to get additional information on the structure of the RuPhos Pd G3 in DMSO-*d*_6_ solution (Figure 11). According to the X-ray diffraction data of RuPhos Pd G3 crystallized from methanol, there was a Pd—C=C bond in the molecules, while the OMes^−^ ion was located in the outer coordination sphere (Figure 5). In such a case, one of the O-*i*-Pr groups of RuPhos was located relatively close to the proton at 7.26 ppm of the aminobiphenyl fragment. It could be expected that such a structure in solution would give rise to a correlation of the CH proton of the O-*i*-Pr group with the proton of the aminobiphenyl fragment, while the coordination of the OMes^−^ anion to the Pd^2+^ ion instead of the C=C bond would increase the separation between the iPr group and aminobiphenyl fragment and break the correlation (Figure 5). Indeed, it was shown by the 1D ROESY experiment (Figure 11) that the proton of the O-*i*-Pr group at 4.50 ppm (CH proton of the O-*i*-Pr group) correlated with the proton of the aminobiphenyl fragment at 7.26 ppm, suggesting the spatial proximity of these protons, which is the argument for the presence of the RuPhos Pd G3 isomer with a Pd—C=C coordination bond in DMSO-*d*_6_ solution.

It was shown by ^1^H-^31^P-HMBC, both for RuPhos Pd G3 and XPhos Pd G3 in DMSO-*d*_6_, that protons of the cyclohexane and aromatic nuclei had correlations with the phosphorus atom (Figure 12 and Appendix A). In the case of RuPhos Pd G3, there are six intense cross peaks of the ^31^P atom with protons at 7.76, 7.44, 7.50, 6.73, 7.04 and 6.94 ppm in the aromatic region of ^1^H-^31^P HMBC and in the ^1^H-^31^P CIGAR-HMBC spectra, which correspond to the J (^1^H-^31^P) coupling constants (Figure 12, Appendix A, Appendix A). Two additional cross peaks of ABP protons at 7.29 and 7.18 ppm with P atoms appeared in the ^1^H-^31^P CIGAR-HMBC spectra, which were not observed in ^1^H-^31^P HMBC (Appendix A, Appendix A). Correlations of ^31^P of the RuPhos fragment of RuPhos Pd G3 with two protons of the amino group at 7.04 ppm and the proton of the phenyl ring at 6.94 ppm of the aminobiphenyl fragment are the most informative and point to the presence of a coordination bond between the Pd^2+^ ion and phosphorus atom, also indicating that dissociation of the complex in solution did not occur (Figure 12). In addition, no extraneous signals were found in the ^31^P and ^1^H spectra, which also reinforces the latter claim.

There are four cross peaks of protons at 7.93, 7.56, 6.97 and 7.26 ppm in the ^1^H-^31^P HMBC and ^1^H-^31^P CIGAR-HMBC spectra of XPhos Pd G3 in DMSO-*d*_6_ solution (Appendix A, Appendix A). The correlation of a phosphorus atom with the proton at 7.26 ppm of the amino group of the ABP fragment points to the absence of the dissociation of the XPhos Pd G3 in DMSO-*d*_6_.

We did not find correlations of the ^31^P and H atoms of the isoprolyl- or isoproposy-substituted phenyl rings of the Phos ligands. The formation of a Pd-C=C bond in Phos Pd G3 complexes had to favor to such correlations, and they could be additional evidence for the existence of such isomers in solutions. Therefore, the shifts of ^31^P signals (Figure 9) are the sole evidence for the existence of different isomers of Phos Pd G3 in solutions.

We can note that single crystals of **5** (containing a Pd-C=C bond) were obtained from methanol; a similar isomer of **3** (compound **3b**) was crystallized from acetone. In CDCl_3_, RuPhos Pd G3 also exists the form which has a Pd-C=C bond, and the same isomer of XPhos Pd G3 exists in DMSO-*d*_6_. Another isomer of **3** (compound **3a**) was crystallized from ethyl acetate. From the results of NMR spectral studies, the mixture of isomers XPhos Pd G3 exists in CDCl_3_ solution (vide infra). It seems that the high dielectric constant of the solvent (ε) favors the dissociation of mesylate and Pd(ABP)(Phos)^+^cation and thus favors the formation of the isomer that has a Pd-C=C bond (ε for CHCl_3_, ethylacetate, acetone and DMSO are 5, 6, 20 and 47, respectively, and the form with coordinated methylate was detected in the case of the first two solvents, while the latter two contained only the isomer with a Pd-C=C bond).

### 2.3. ^1^H-NMR Spectroscopy for Assessment of the Content of the Impurities

The signals of impurities may superimpose on the bands of the main substance, increasing their intensities above the theoretical values (Appendix A). For the sake of convenience, the multiplets which are discussed below are denoted by Latin letters A–G (Figure 13). Signals A, H and I belong to the Phos Pd G3 complexes (vide infra), while signals **B**–**G** belong to the impurities, and the positions of the latter group of signals are the same in the case of Xphos Pd G3 and RuPhos Pd G3.

A comparison of the spectra of synthesized Phos Pd G3 complexes with the spectra of the precursors [11,20,21] and solvents used in the syntheses [22] showed that the samples, typically prepared in our research work, were mainly contaminated with **1**, THF and hexane. The presence of **1** in a sample increases the intensity of multiplet B in the aromatic region (Figure 14) and D in the aliphatic, whereas the presence of hexane intensifies multiplets F (CH_2_) and G (CH_3_). The admixture of THF gives rise to a new multiplet C at 3.6 ppm (OCH_2_) and causes the intensification of multiplet E (CH_2_).

Since the intensity of the NMR signal is directly proportional to the number of resonating nuclei, the content of the corresponding impurities in the sample of the Phos Pd G3 complex can be determined from Equation (1):*I_exp_* = *I*_theor_ + *nh*(1)
where *I_exp_* and *I*_theor_ are the experimental and theoretical intensities of signals B, C or F (vide infra); *n* is the content of impurity per 1 mol of the Phos Pd G3 complex and *h* is the number of protons of impurity contributing to this signal. If the integrated intensities of signals are normalized to the most weak-field signal A (*I*_A_ = 1, see Figure 14), the amounts of impurities (mole per mole) can be expressed as shown by Equations (2)–(4):*n*([Pd(ABP)(OMs)]_2_) = (*I*_B_ − 5)/8(2)
*n*(THF) = *I*_C_/4(3)
*n*(hexane) = (*I*_F_ − *a*)/8(4)
where *a* = 6 (XPhos) or 15 (RuPhos). Then, the weight percent of impurity (*w_i_*) is expressed by Equation (5):*w_i_* = *n_i_M_i_*/*m*_total_ × 100%(5)
where *M_i_* is the molar mass of impurity (739.4, 72.1 and 86.2 g/mol for **1**, THF and hexane, respectively), *M*_complex_ is the molar mass of the Phos Pd G3 complex (846.5 and 836.4 g/mol for XPhos Pd G3 and RuPhos Pd G3, respectively), *m*_total_ = *M*_complex_ + ∑*n_i_M_i_* and the weight percent of Phos Pd G3 (*w*_complex_) can be calculated using Equation (6):*w*_complex_ = *M*_complex_*/m*_total_ × 100%(6)

The signal A is used only for calibration of the intensities of signals B, C and F (I_A_ = 1).

In order to verify the proposed method, the spectra of solutions containing a mixture of Phos Pd G3 and a known quantity of specially added impurity **1** were studied (Figure 15). As expected, a close to linear correlation was observed between the intensity of multiplet **B** and the molar ratio of the components (Figure 15).

To assess the error in determining the intensity of the signals in the ^1^H-NMR spectrum during routine measurements, the spectra of 20 samples of XPhos Pd G3 produced in different syntheses were recorded. Multiplets H and I (see Figure 13a) were considered because they did not overlap with the impurity signals, so their integrated intensities were expected to be constant from sample to sample. The root mean square (RMS) deviations of their intensities was found to be 0.09/0.43 with respect to the average values of 0.97/8.18. Therefore, the average dispersion of the intensity, which may be calculated as a ratio of RMS deviation to the theoretical value of integrated intensity (*I*_theor_ = 1 and 8, respectively), is 9.0 (H) or 5.3% (I). These values provide a rough estimate of the relative accuracy of the proposed method.

### 2.4. ^31^P-NMR Spectra

^31^P-NMR spectroscopy is widely used for the QC of phosphorus compounds, and it is convenient method to monitor the completeness of PdG3(Phos) formation in the reaction mixture, which is achieved with the disappearance of free phosphine signal and the appearance of the complex signal downfield. The analysis of ^31^P-NMR spectra also allows us to check the presence of other phosphorus-containing impurities.

In order to test the feasibility of this technique for the quality control of XPhos Pd G3 and PdG3(RuPhos), the spectra of dialkylbiaryl phosphines and their mixtures with **1** in a stochiometric ratio (2:1) were studied in solvents which are usually used for synthesis of precatalysts (THF or DCM) [11]. For comparison, chloroform was also considered, and it was found that for both chlorinated solvents, chemical shifts differ by less than 1 ppm. The values of ^31^P chemical shifts together with the literature data for phosphines and corresponding PdG3(Phos) are listed in Table 5.

In the spectra of pure ligands, for XPhos and RuPhos dissolved in CHCl_3_ or CH_2_Cl_2_, only one signal is present in the high field, and its chemical shift (δ ≈ −10 ppm) is in agreement with literature value [20,21]. The same signal is observed in the spectra measured in THF, which was freshly distilled and purged with argon. On the other hand, in commercial THF which was not specially purified, an additional downfield signal appears at ~40 ppm (Figure 16).

A similar distinction is observed for the stoichiometric mixtures of **1** and the phosphine ligand. The spectra measured in chlorinated solvents or purified THF exhibit one resonance at ~35 (**1** + XPhos) or ~40 ppm (**1** + RuPhos), which is in agreement with the literature values for the corresponding Phos Pd G3 ([11]. However, when the stoichiometric mixture of **1** and RuPhos is dissolved in commercial THF, two signals (among which the upfield one at 39 ppm is attributed to RuPhos Pd G3) are present in the ^31^P-NMR spectrum.

The appearance of the additional downfield signals in nonpurified THF is probably due to the oxidation of phosphines by peroxide impurity in this solvent. The chemical shifts of these signals are in the typical range for ^31^P-NMR resonances of dialkylbiaryl phosphine oxides [23,24]. The RuPhos-based phosphine oxide was also identified in the filtrate remaining after the separation of RuPhos Pd G3 from the reaction mixture: ca. 3% mol. Of the RuPhos was converted to the corresponding phosphine oxide (compound **5**), which was isolated and characterized by ^1^H- and ^31^P-NMR spectroscopy (Figure 17).

However, from monitoring of the reaction progress by ^31^P-NMR spectroscopy, it was found that RuPhos was not oxidized during the synthesis of RuPhos Pd G3 in freshly distilled THF (several hours). Therefore, the occurrence of phosphine oxide in the filtrate after the preparative synthesis of RuPhos Pd G3 can be caused by the oxidation of the palladium complex with atmospheric oxygen during isolation of the RuPhos Pd G3 precipitate from the reaction mixture on air. It was also found that individual RuPhos was stable in freshly purified THF for at least 3 days since no other signals appeared in the ^31^P-NMR spectra, whereas in the spectra of RuPhos Pd G3 dissolved in THF under argon atmosphere, several signals at 48.46, 41.88 and 36.04 ppm appeared after 3-day storage in the sealed NMR tube. It can be concluded that RuPhos bound to the Pd^2+^ ion is oxidized more easily compared to noncoordinated RuPhos.

In the case of XPhos Pd G3, the downfield signal at ~64 ppm (along with the signal at 36 ppm) is present not only in unpurified THF but also in the solvents which usually do not form peroxides, such as acetone or ethyl acetate (Figure 8)**.** Based on the results of X-ray characterization of **3a** and **3b**, the upfield resonance in the ^31^P-NMR spectra at 36 ppm can be attributed to the cationic [PdG3(XPhos)]^+^ species, whereas the downfield resonance at 64 ppm can be attributed to neutral PdG3(XPhos).

Therefore, the quality control of XPhos Pd G3 and RuPhos Pd G3 using ^31^P-NMR spectroscopy should be carried out in solvents which do not form peroxides during storage (such as chloroform or DCM); otherwise, the phosphine oxide impurity may form directly in an NMR tube even if it is absent in the studied sample. Synthesis of the precatalysts also should be performed in freshly distilled THF to reduce the probability of their contamination with oxidized byproducts.

## 3. Experimental

### 3.1. Materials and Methods

The starting materials (Pd(OAc)_2_, 2-aminobiphenyl, methanesulfonic acid, XPhos, RuPhos) were obtained from commercial sources (Enamine Ltd., Kyiv, Ukraine) and used as received. 2-ammoniumbiphenyl mesylate was synthesized according to the published procedure [11]. Toluene and tetrahydrofuran (THF) were dried according to well-known methods [25] and distilled under argon atmosphere. Hexane was dried over P_2_O_5_ and distilled under argon atmosphere. Operations with phosphine-containing compounds were performed in argon atmosphere using the Schlenk technique (including preparation of samples for NMR measurements).

NMR spectra were measured at 500 (^1^H) or 202 MHz (^31^P), and samples were sealed in NMR tubes under Ar. Chemical shifts (δ) and are reported in ppm relative to tetramethylsilane (^1^H) or a 15% solution of phosphoric acid in D_2_O (^31^P). PSYCHE (pure shift yielded by chirp excitation), ^13^CAPT, ^31^P coupled and decoupled proton NMR, HSQC (heteronuclear single quantum correlation), COSY (^1^H-^1^H correlation spectroscopy), ROESY (rotating frame overhauser enhancement spectroscopy), ^1^H-^13^C HMBC (8 Hz) (heteronuclear multiple bond correlation) and ^1^H-^31^P HMBC (8Hz) and ^1^H-^31^P CIGAR (3–12 Hz) are given in Appendix A (frequencies are shown on the respective spectra).

All crystallographic measurements were performed on a Bruker Smart Apex II diffractometer operating in the ω scans mode, and temperature was 173(2) K for all measurements. The intensity data were collected within the θ_max_ ≤ 26.63° using Mo-Kα radi-ation (λ = 0.71078 Å). The intensities of 7606 reflections were collected (1901 unique reflections, Rmerg = 0.0485). The structures were solved by direct methods and refined by the full-matrix least-squares technique in the anisotropic approximation for nonhydrogen atoms using the Bruker SHELXTL program package [26]. All CH hydrogen atoms were placed at calculated positions and refined as the ‘riding’ model. Crystallographic data and structure refinement parameters for **2b**, **3a**, **3b**, **4** and **5** are presented in Table 6. Supplementary crystallographic data for the compounds synthesized are given in CCDC numbers 2,080,959 (**2b**), 2,080,960 (**3a**), 2,080,962 (**3b**), 2,080,964 (**4**) and 2,080,965 (**5**). These data can be obtained free of charge from The Cambridge Crystallographic Data Centre via www.ccdc.cam.ac.uk/data_request/cif).

### 3.2. Synthesis of the Complexes

Samples of dimeric palladacycle [Pd(ABP)(OMs)]_2_ and precatalysts Pd(ABP)(XPhos)(OMs) and Pd(ABP)(RuPhos)(OMs) were prepared by a slightly modified procedure, reported in [11].

#### 3.2.1. Synthesis of [Pd(ABP)(OMs)]_2_ (1) and Isolation of Impurity Pd(ABP)(HABP)(OMs) (as Two Solvates **2a** and **2b**)

Palladium acetate (44.9 g, 0.2 mol, 1 eq.) and 2-ammoniumbiphenyl mesylate (53.0 g, 0.2 mol, 1 eq.) were charged to a 1 L round bottom Schlenk flask containing a magnetic stir bar. The flask was tightly capped, evacuated and backfilled with argon three times. Freshly distilled anhydrous toluene (0.8 L) was added in the flask under argon counterflow. The flask was capped with a rubber septa and heated at 50 °C during 2 h under stirring. After cooling to room temperature the reaction mixture was filtered and the precipitate was washed with toluene (2× 100 mL). The filtrate was separated and the filter cake was washed with methyl *tert*-butyl ether (MTBE, 3× 150 mL) and dried in vacuum at 40 °C for 2 h to obtain 69.2 g of 1 with the composition of 1·0.91C_6_H_5_CH_3_·0.04MTBE·0.03CH_3_COOH (83% yield based on Pd). ^1^H-NMR (500 MHz, DMSO-*d*_6_) δ 7.63–7.57 (m, 1H), 7.53–7.40 (m, 3H), 7.39–7.33 (m, 1H), 7.27–7.12 (m, 5H), 2.39 (s, 3H). This solvate was used directly for synthesis of 3 and 4 without additional purification or drying.

The separated filtrate was evaporated, and the residue was treated with MTBE using ultrasound and then proceeded in the same way as the main part of **1**, yielding 0.95 g of a dark brown solid with the composition of **1**·0.06C_6_H_5_CH_3_·0.12MTBE.

White precipitate was formed in the filtrate remaining after the isolation of the second crop of the dimer **1**. The precipitate was filtered, washed with CHCl_3_ and dried to obtain 50 mg of complex Pd(ABP)(HABP)(OMs) (**2a**) with the composition of Pd(ABP)(HABP)(OMs)·0.12C_6_H_5_CH_3_·0.33MTBE. ^1^H-NMR (500 MHz, DMSO-*d*_6_) δ 7.62–7.56 (m, 1H), 7.49–7.36 (m, 6H), 7.34–7.28 (m, 2H), 7.26–7.10 (m, 4H), 7.03 (t, *J* = 7.7 Hz, 1H), 6.97 (d, *J* = 7.5 Hz, 1H), 6.76 (d, *J* = 8.0 Hz, 1H), 6.64 (t, *J* = 7.4 Hz, 1H), 3.06 (s, CH_3_O of MTBE), 2.43 (s, 3H), 2.28 (s, CH_3_ of toluene), 1.09 (s, CH_3_ of MTBE).

The crystallization of **2a** from acetonitrile led to the formation of solvate [Pd(ABP)(HABP)(CH_3_CN)](OMs)·H_2_O (**2b**), which was analyzed by single crystal X-ray diffraction.

#### 3.2.2. Synthesis of Pd(ABP)(XPhos)(OMs) (Abbreviated as XPhos Pd G3, (**3**) and PdCl_2_(XPhos)_2_·CHCl_3_ (**4**))

A two-necked 2 L round bottom Schlenk flask equipped with a magnetic stir bar was charged with **1** (53.6 g, 64 mmol, 0.5 eq.) and XPhos (61.5 g, 129 mmol, 1 eq.). The flask was capped, three times evacuated and backfilled with argon. Anhydrous and deoxygenated tetrahydrofuran (1.4 L) were added in the flask under argon counterflow. The reaction mixture was stirred at room temperature for 1.5 h and then it was filtered under argon atmosphere. About 1.2 L of the filtrate was evaporated under vacuum via a rotor evaporator. Deoxygenated hexane (1.2 L) was added to the dark brown residue and the formation of a beige precipitate was observed. The flask was placed in an ultrasonic bath and sonicated for 20 min. A beige precipitate was filtered, washed with hexane (3× 250 mL) and dried under vacuum at 50 °C for 1 h to obtain 92.0 g of **3** with the composition of Pd(ABP)(XPhos)(Oms) ·0.18THF (83% yield based on Pd). ^1^H-NMR of **3** (500 MHz, DMSO-*d*_6_) δ, ppm: 7.95 (m, 1H), 7.65 (s, 1H), 7.61–7.56 (m, 2H), 7.51 (s, 2H), 7.30–7.20 (m, 6H), 7.01–6.93 (m, 3H), 3.60 (m, CH_2_O of THF), 3.30 (quint, *J* = 6.92 Hz, 1H), 2.77 (quint, *J* = 6.75 Hz, 1H), 2.41–2.33 (m, 2H), 2.29 (s, 3H), 2.12 (m, 1H), 2.02 (m, 1H), 1.93–1.83 (m, 2H), 1.78–1.64 (m, 6H), 1.59–1.39 (m, 8H), 1.34–1.04 (m, 13 H), 0.98 (d, *J* = 6.3 Hz, 3H), 0.95–0.89 (m, 2H), 0.87–0.82 (m, 8H), 0.73 (d, *J* = 6.4 Hz, 4H), 0.01 (m, 1H).

The filtrate remaining after isolation of the main part of **3** was treated with methanol, filtered and crystallized from the CH_3_OH/CHCl_3_ mixture leading to precipitation of 30 mg of complex PdCl_2_(XPhos)_2_·CHCl_3_ (**4)** as yellow crystals, which were collected and analyzed by NMR and single crystal X-ray diffraction. ^1^H-NMR of the bulk sample was used for obtaining single crystals of **4** (400 MHz, DMSO-*d*_6_) δ, ppm: 8.3 (s, br, 2H), 7.37 (t, *J* = 7.3 Hz, 2H), 7.29 (t, *J* = 7.3 Hz, 2H), 7.01 (s, 5H), 6.97 (s, 1H), 2.91 (quint, *J* = 6.8 Hz, 4H), 1.5 (m, br, 12H), 1.28 (d, *J* = 7.0 Hz, 12H), 1.20 (d, *J* = 5,8 Hz, 12H). ^31^P (202 MHz, DMSO-*d*_6_) δ, ppm: 45.63.

Single crystals of two isomers of **3**, **3a** and **3b** were obtained by crystallization of the bulk sample Pd(ABP)(XPhos)(OMs)·0.18THF from pure ethyl acetate or from acetone containing ca. 0.5% of THF (added to increase solubility of the complex), respectively.

#### 3.2.3. Synthesis of Pd(ABP)(RuPhos)(OMs) (Abbreviated as RuPhos Pd G3, **5**)

Dimer **1** (11.715 g, 14.1 mmol, 0.5 eq.) and RuPhos (13.139 g, 28.2 mmol, 1 eq.) were charged to a 250 mL round bottom Schlenk flask equipped with a magnetic stir bar. The flask was capped with a rubber septa, three times evacuated and backfilled with argon. THF (115 mL) was added to the flask via stainless steel cannula. The reaction mixture was stirred at room temperature for 2 h and formation of precipitate was observed. The mixture was stirred for another 1 h and filtered. The filtrate was separated. The filter cake was washed with hexane (3× 80 mL) and dried in vacuum at 40 °C for 1 h to obtain 13.55 g of **5** with the composition of Pd(ABP)(RuPhos)(OMs)·0.87THF as a light yellow solid.

To increase the product yield, the filtrate was evaporated under vacuum at a rotor evaporator until about 90% of solvent was removed. Hexane (100 mL) was added to the dark brown residue under argon counterflow. The content of the flask was alternately triturated with hexane using ultrasonic irradiation and intense magnetic stirring. The obtained light brown precipitate was filtered, washed with hexane (3× 100 mL) and dried under vacuum at 40 °C for 1 h yielding 13.82 g of tan solid. The product was purified using column chromatography on silica gel in mixture of CHCl_3_/CH_3_OH (100:0 →80:20) as eluent. Fractions containing **5** were evaporated and triturated with hexane to afford a light beige precipitate that was washed with hexane (3× 30 mL) and dried in vacuum at 40 °C for 1 h to obtain 4.50 g of **5** with the composition of Pd(ABP)(RuPhos)(OMs)·0.56CHCl_3_. The total yield of the first and the second crops of **5** was 70%. Single crystals of **5** were obtained from methanol.

A filtrate collected after separation of the second crop of **5** was evaporated and treated with methanol. The resulting light yellow solid was filtered, washed with methanol and dried to obtain 0.42 g of light yellow solid **6**, which was identified as 2-dicyclohexylphosphinoxide-2′,6′-diisopropoxybiphenyl (phosphine oxide of RuPhos). ^1^H-NMR (500 MHz, CDCl_3_) δ 8.01 (q, *J* = 7.3 Hz, 1H), 7.31–7.13 (m, 3H), 6.82 (d, *J* = 6.9 Hz, 1H), 6.49 (d, *J* = 8.2 Hz, 2H), 4.35 (hept, *J* = 6.3 Hz, 2H), 2.26 (d, *J* = 12.0 Hz, 2H), 1.93 (s, 4H), 1.70–1.62 (m, 2H), 1.58–1.46 (m, 8H), 1.19 (d, *J* = 5.9 Hz, 6H), 1.03 (p, *J* = 9.6 Hz, 10H), 0.84 (t, *J* = 10.5 Hz, 2H). ^31^P-NMR (202 MHz, CDCl_3_) δ 47 ppm.

## 4. Conclusions

In the present study, it was demonstrated that Phos Pd G3 precatalysts could undergo isomerization in solution and existed in two forms, which were distinguished by the presence of mesylate or a C=C bond in the coordination sphere of the Pd^2+^ ion. These forms could be isolated as pure solid compounds, which was confirmed by single crystal X-ray diffraction. It was shown that NMR spectroscopy could be used as a rapid method for quality control of Buchwald G3 precatalysts with XPhos and RuPhos. Assignment of the chemical shifts of ^1^H signals of the coordinated RuPhos and ABP^−^ fragment in RuPhos Pd G3 and some signals in the NMR spectra of XPhos Pd G3 in DMSO-*d*_6_ solution was carried out. The intensities of the ^1^H-NMR signals of the palladium precatalyst and impurities were shown to be additive; thus, the percentage of a certain impurity in a sample could be easily calculated on the basis of the integrated intensity of selected signals in its spectrum. The relative error of the proposed technique was assessed to be within 10%. ^31^P-NMR spectroscopy could also be used to verify the purity of the precatalyst with regard to dialkylbiaryl phosphine admixture; however, care must be taken to avoid possible oxidation of the sample compound by organic peroxides.

We believe that the results of this study will be useful for routine express quality control of Phos Pd G3 precatalysts, especially for the control of captured solvent content in the samples. Such quality control can be a good alternative to desolvation of the precatalysts for their use in fine organic synthesis.

## Figures and Tables

**Figure 1 molecules-26-03507-f001:**
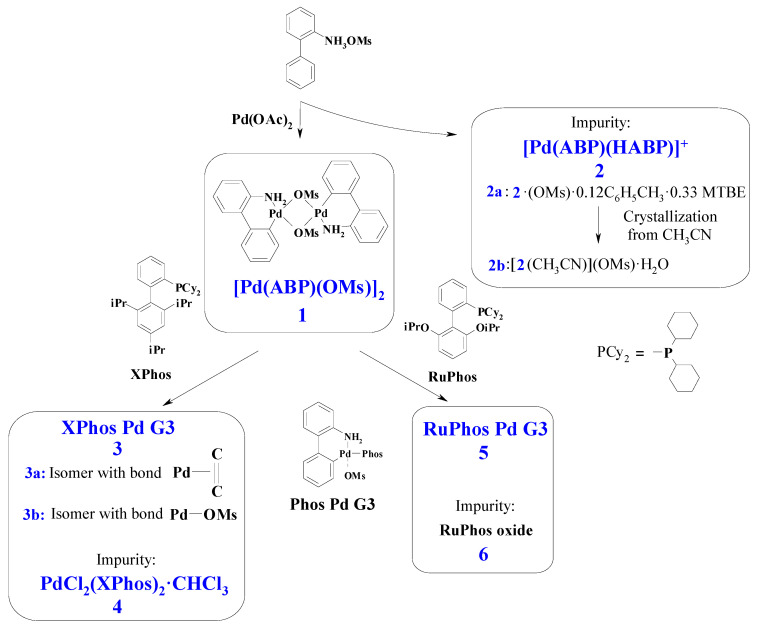
General scheme for synthesis of Buchwald G3 precatalysts and other compounds considered in this paper. Only distinctive features of **3a** and **3b** are indicated on the scheme. Phos Pd G3 is the general formula for the complexes **3** and **5** (for **3** Phos = Xphos, for **5** Phos = RuPhos). See text for details.

**Figure 2 molecules-26-03507-f002:**
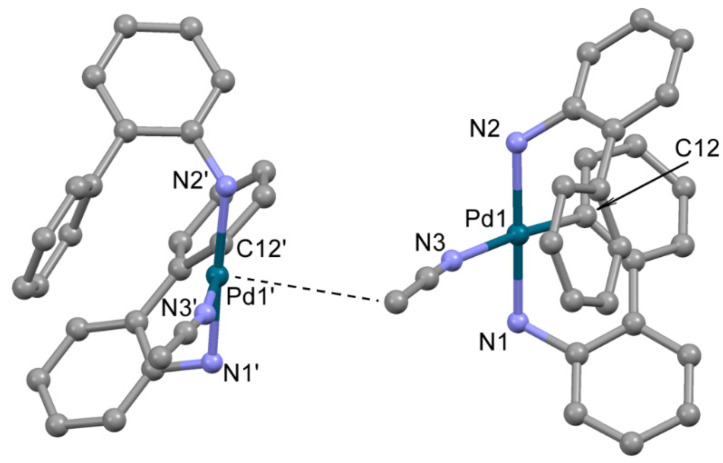
The molecular structure of **2b**. Two neighboring cations are shown, Pd---CH3 contact is shown by dotted line. Noncoordinated mesylate anions and water molecules are not shown. Thermal ellipsoids are shown at 0.15 probability level. H atoms are omitted for clarity.

**Figure 3 molecules-26-03507-f003:**
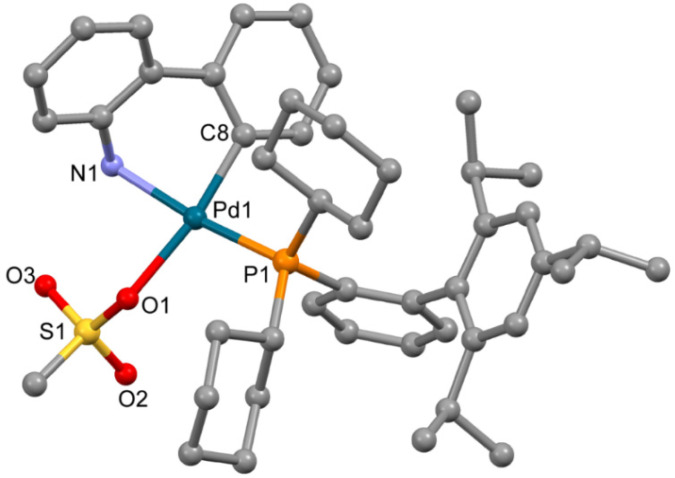
Molecular structure of **3a**. Noncoordinated ethylacetate is now shown. Thermal ellipsoids are shown at 0.15 probability level. H atoms are omitted for clarity.

**Figure 4 molecules-26-03507-f004:**
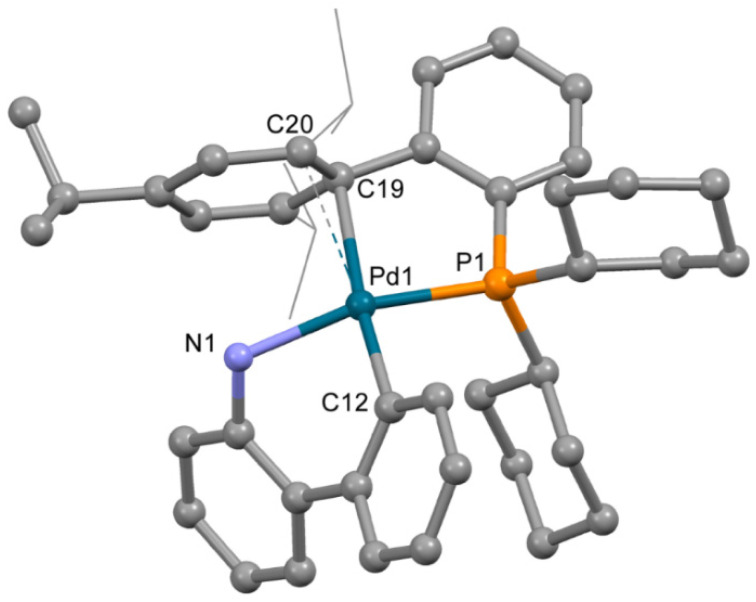
Molecular structure of **3b**. Noncoordinated mesylate anion is now shown. Thermal ellipsoids are shown at 0.15 probability level. H atoms are omitted for clarity. Isopropyl substituents in Xphos are shown in wireframe style for clarity.

**Figure 5 molecules-26-03507-f005:**
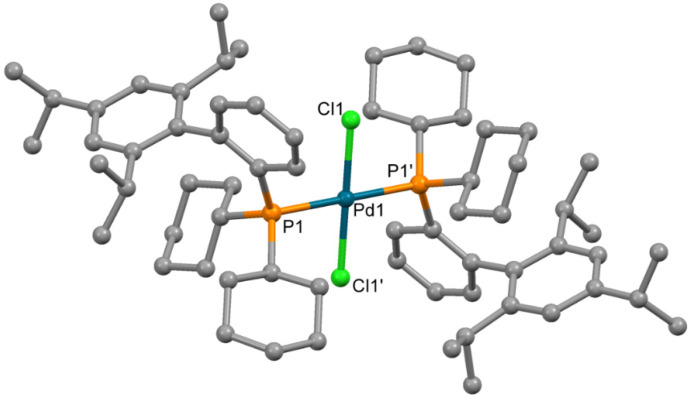
Molecular structure of **4**. Non-coordinated chloroform is not shown Thermal ellipsoids are shown at 0.15 probability level. H atoms are omitted for clarity.

**Figure 6 molecules-26-03507-f006:**
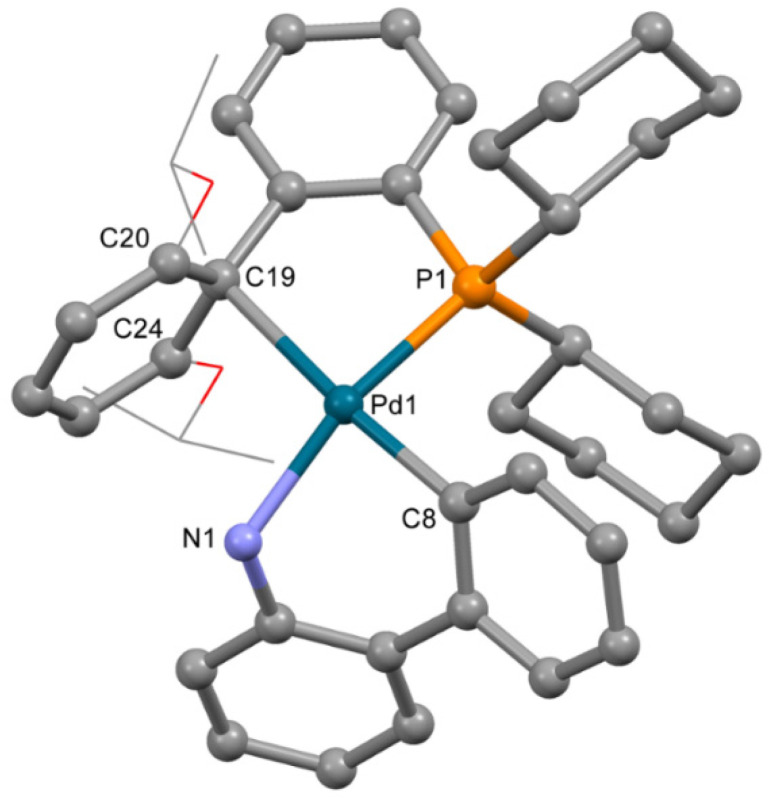
Molecular structure of **5**. Noncoordinated mesylate anion is now shown. Thermal ellipsoids are shown at 0.15 probability level. H atoms are omitted for clarity. Isopropoxy substituents in RuPhos are shown in wireframe style for clarity.

**Figure 7 molecules-26-03507-f007:**
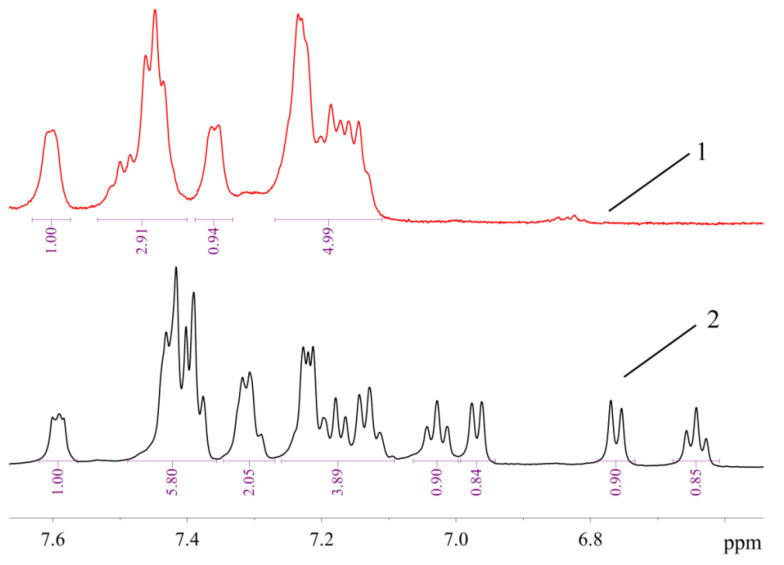
The ^1^H-NMR spectra of **1** (1) and **2a** (2) in DMSO-*d*_6_ (region of H atoms of aromatic systems).

**Figure 8 molecules-26-03507-f008:**
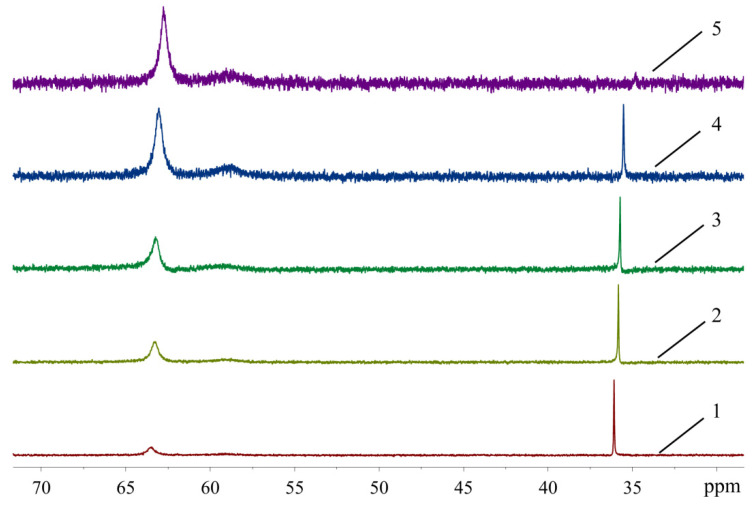
The ^31^P-NMR spectra of XPhos Pd G3 in acetone (1), acetone:ethyl acetate = 2:1 (2), acetone:ethyl acetate = 1:1 (3), acetone:ethyl acetate = 1:2 (4) and ethyl acetate (5).

**Figure 9 molecules-26-03507-f009:**
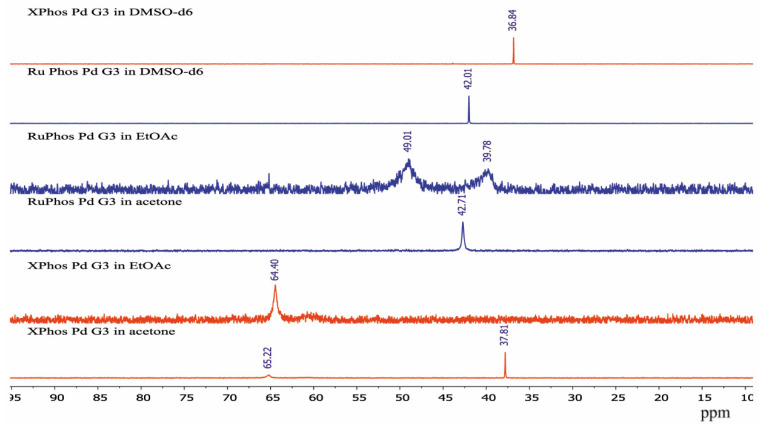
The ^31^P-NMR spectra of XPhos Pd G3 and RuPhos Pd G3 in acetone, ethyl acetate solution and DMSO-*d*_6_ solutions.

**Figure 10 molecules-26-03507-f010:**
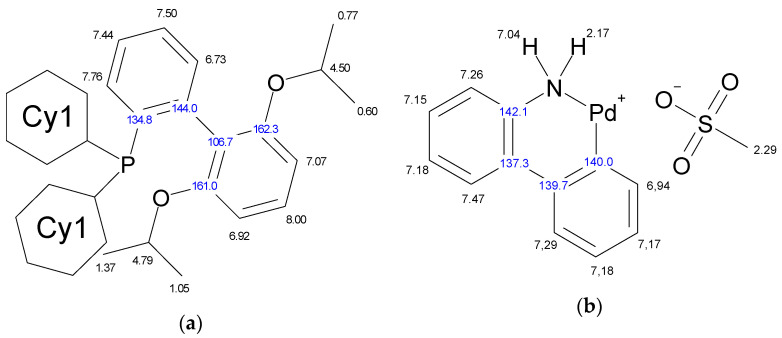
Assignment of the signals specified by their chemical shifts of ^1^H (black) and ^13^C (blue), of coordinated RuPhos in RuPhos Pd G3 (**a**), ABP^−^ fragment in RuPhos Pd G3 (**b**), coordinated XPhos in XPhos Pd G3 (**c**) and NH_2_-group of ABP in XPhos Pd G3 (**d**), based on 1D and 2D NMR experiments in DMSO-*d*_6_.

**Figure 11 molecules-26-03507-f011:**
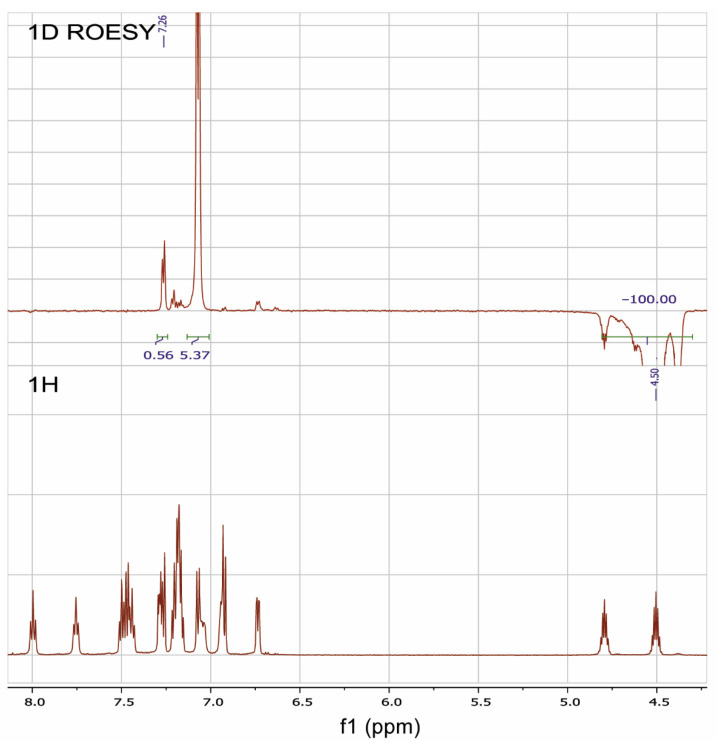
^1^H-NMR and 1D ROESY spectra of RuPhosPdG3 at saturating of the proton of one of the O-*i*-Pr groups at 4.50 ppm.

**Figure 12 molecules-26-03507-f012:**
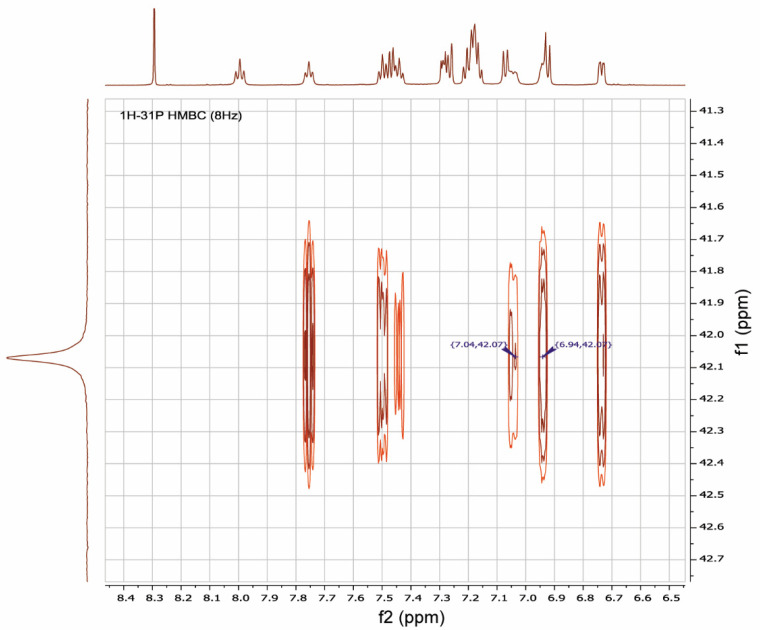
A fragment of ^1^H-^31^P HMBC spectrum illustrating the key correlations of protons at 7.04 and 6.94 ppm of aminobiphenyl fragment with a phosphorus atom of RuPhos in RuPhos Pd G3 in DMSO-*d*_6_ solution.

**Figure 13 molecules-26-03507-f013:**
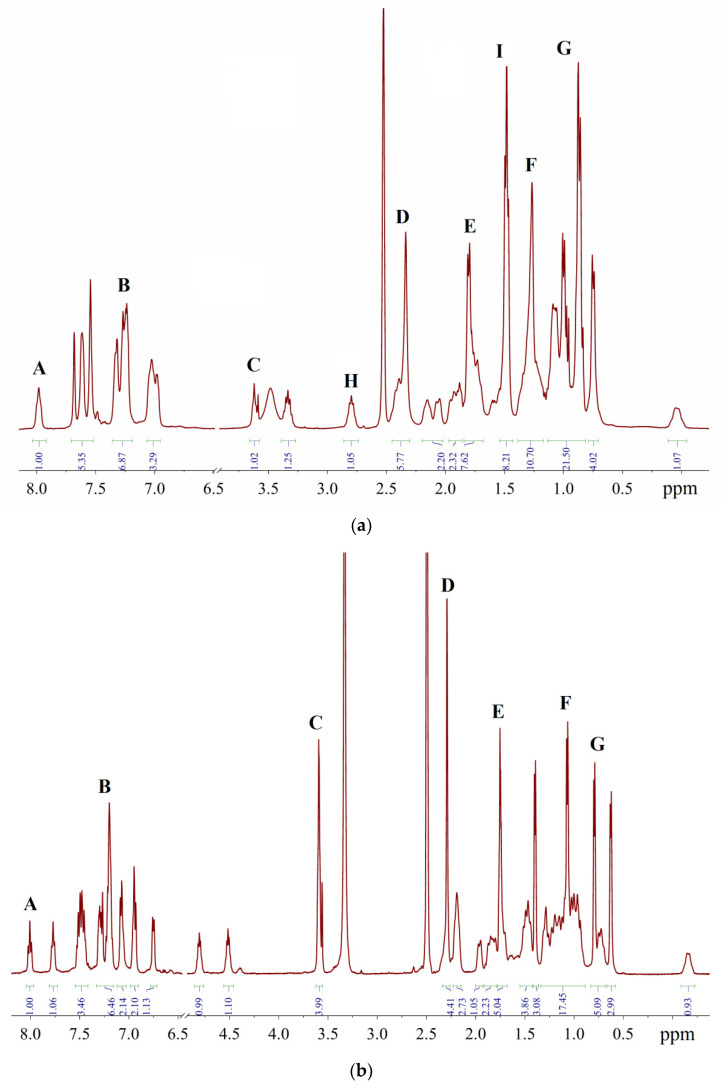
Representative ^1^H-NMR spectra in DMSO-*d*_6_: (**a**)—sample of XPhos Pd G3 containing 15.6% of **1**, 1.3% of THF and 4.9% of hexane; (**b**)—sample of RuPhos Pd G3 containing 14.1% of **1**, 6.8% of THF and 2.5% of hexane.

**Figure 14 molecules-26-03507-f014:**
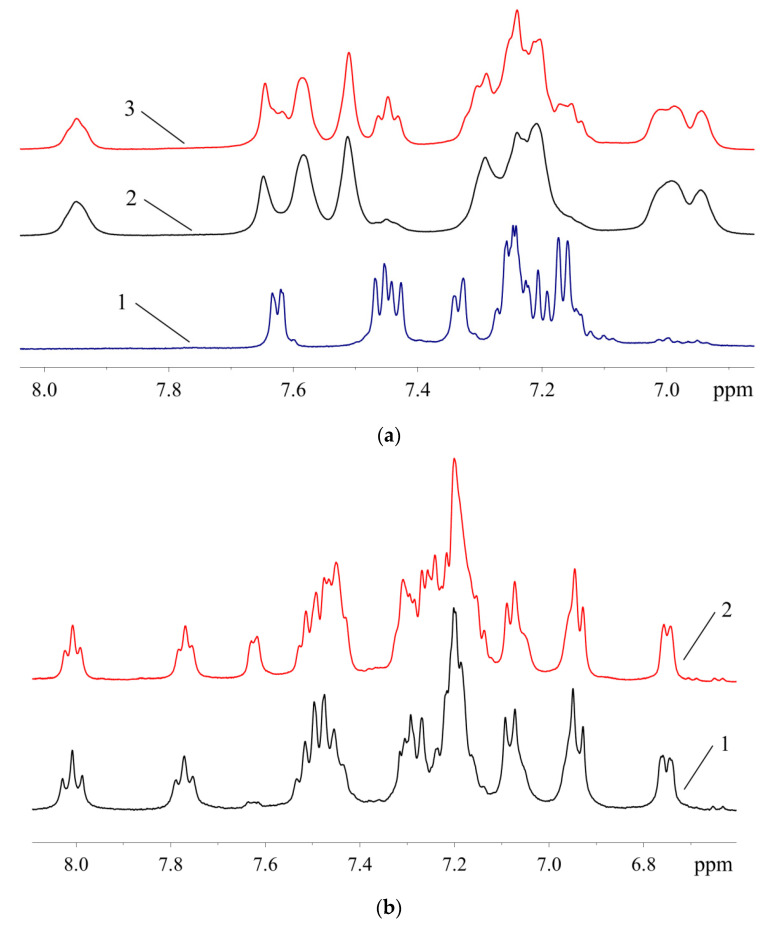
Aromatic region of the ^1^H-NMR spectra: (**a**)—**1** (1), samples of XPhos Pd G3 containing 8.5% (2) and 26.4% (3) of **1**; (**b**)—samples of RuPhos Pd G3 containing 9.3% (1) and 29.0% (2) of **1**.

**Figure 15 molecules-26-03507-f015:**
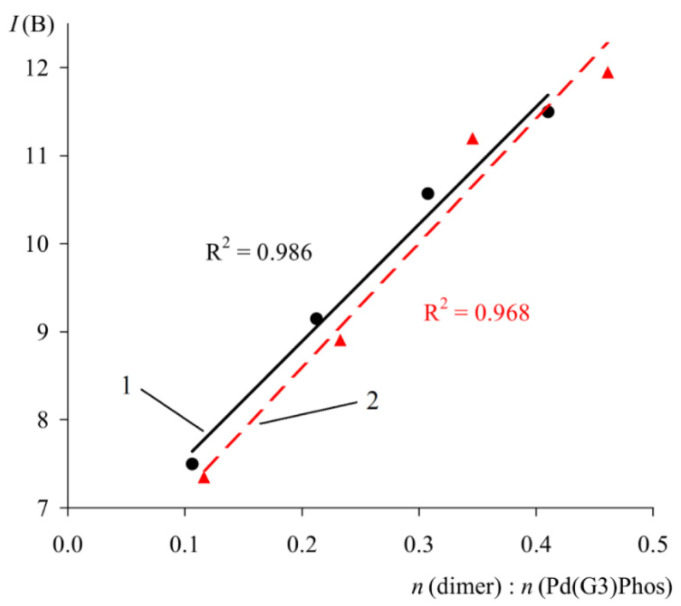
Plot of *I*_B_ versus molar ratio of **1** to XPhos Pd G3 (1) and RuPhos Pd G3 (2). *C*(PdG3(Phos)) = 40 mmol/L.

**Figure 16 molecules-26-03507-f016:**
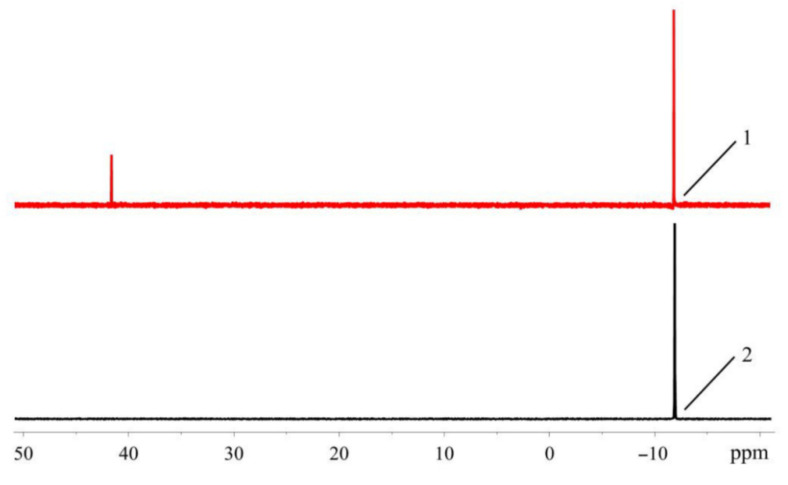
^31^P-NMR spectra of XPhos in unpurified commercial (1) and freshly distilled THF (2).

**Figure 17 molecules-26-03507-f017:**
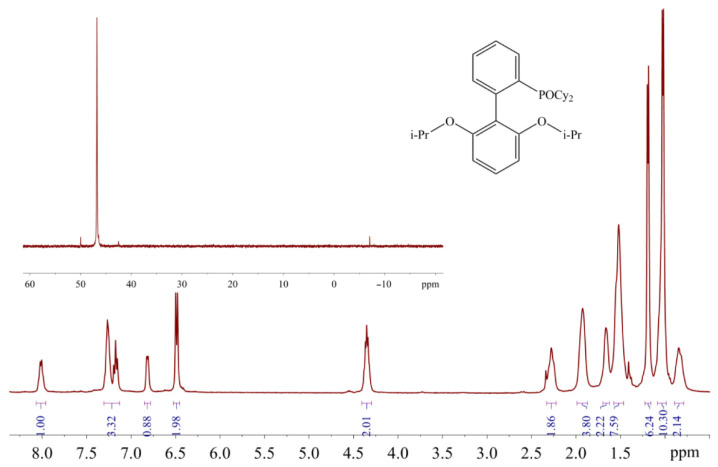
^1^H and ^31^P (in the inset) NMR spectra of **5** in CDCl_3_.

**Table 1 molecules-26-03507-t001:** Selected bond length and angles in **2b**.

Bonds, Å		Angles, Deg	
Pd1–N1	2.042(3)	N1–Pd1–C12	84.40(15)
Pd1–N2	2.092(3)	N1–Pd1–N3	92.82(13)
Pd1–N3	2.118(3)	N2–Pd1–N3	88.13(13)
Pd1–C12	1.978(4)	N2–Pd1–C12	94.79(15)

**Table 2 molecules-26-03507-t002:** Selected bond length and angles in **3a** and **3b**.

3a		3b	
Bonds, Å			
Pd1–N1	2.128(4)	Pd1–N1	2.097(7)
Pd1–C8	2.008(5)	Pd1–C8	2.005(9)
Pd1–P1	2.2827(14)	Pd1–P1	2.265(2)
Pd1–O1	2.184(3)	Pd1–C19	2.472(8)
		Pd1–C20	2.595(9)
Angles, deg			
C8-Pd1-N1	83.50(18)	C12-Pd1-N1	82.0(3)
N1-Pd1-O1	87.24(14)	C12-Pd1-P1	91.0(3)
C8-Pd1-P1	91.85(15)	N1-Pd1-C19	103.0(3)
O1-Pd1-P1	97.16(10)	P1-Pd1-C19	83.7(2)
		N1-Pd1-C20	91.9(3)
		P1-Pd1-C20	101.6(2)

**Table 3 molecules-26-03507-t003:** Selected bond length and angles in **4**.

Bonds, Å		Angles, Deg	
Pd1–Cl1	2.2975(14)	Cl1–Pd1–P1	87.06(5)
Pd1–P1	2.3430(14)	Cl1–Pd1–P1’	92.94(5)

**Table 4 molecules-26-03507-t004:** Selected bond length and angles in **5**.

Bonds, Å		Angles, Deg	
Pd1–C8	2.007(4)	C8–Pd1–N1	82.19(15)
Pd1–N1	2.120(3)	C8–Pd1–P1	90.14(12)
Pd1–P1	2.2578(11)	N1–Pd1–C19	102.81(13)
Pd1–C19	2.421(4)	P1–Pd1–C19	84.16(10)

**Table 5 molecules-26-03507-t005:** ^31^P chemical shifts (ppm) of XPhos, RuPhos and their mixtures with **1**.

	THF *	CHCl_3_ or CH_2_Cl_2_	Literature Data
XPhos	−12.2/−12.2, 41.2	−12.7	−11.5 (C_6_D_6_) [20]
RuPhos	−10.2/−9.9, 42.7	−10.0	−8.8 (C_6_D_6_) [21]
**1** + XPhos	35.5/64.4	35.8	35.9, 65.2 ** (CDCl_3_) [11]
**1** + RuPhos	39.5/38.5, 48.7	40.3	41.6 (CDCl_3_) [11]

* Freshly distilled/commercial. ** Minor signal.

**Table 6 molecules-26-03507-t006:** Crystallographic data and structure refinement parameters for **1**–**4**.

	2b	3a	3b	4	5
Empirical formula	C_27_H_29_N_3_O_4_PdS	C_50_H_70_NO_5_PPdS	C_49_H_68_NO_4_PPdS	C_68_H_100_Cl_8_P_2_Pd	C_43_H_56_NO_5_ PPdS
Formula weight (g·mol^−1^)	597.99	934.50	904.47	1369.41	836.31
Crystal system	monoclinic	triclinic	monoclinic	triclinic	triclinic
Space group	P 2_1_/n	P-1	P 2_1_/n	P-1	P-1
*a* (Å)	9.9769(2)	12.3572(11)	10.9570(15)	9.6024(6)	10.7681(6)
*b* (Å)	26.9162(7)	13.6870(13)	31.926(4)	13.4732(8)	11.9967(6)
*c* (Å)	10.0825(3)	14.6964(13)	13.281(2)	15.0755(9)	16.2496(8)
α (^o^)	90	101.015(6)	90	89.386(4)	83.623(3)
β (^o^)	106.5480(10)	94.793(6)	99.011(10)	72.425(4)	87.030(4)
γ (^o^)	90	99.427(5)	90	72.356(4)	77.853(3)
*V* (Å^3^)	2595.41(12)	2389.9(4)	4588.5(11)	1764.86(19)	2038.58(18)
*Z*	4	2	4	1	2
*D*_calc_ (g·cm^−3^)	1.530	1.299	1.309	1.288	1.362
μ (mm^−1^)	0.834	0.511	0.528	0.649	0.590
*F*(000)	1224	988	1912	720	876
θ range for data collection (^o^)	2.239 to 26.530	2.063 to 25.491	1.987 to 25.409	2.039 to 25.888	1.935 to 25.440
Reflections collected	28056	28893	27071	24047	26539
Reflections unique	5354	8809	8418	6769	7488
*R* _int_	0.0370	0.0895	0.2107	0.0632	0.0863
Parameters	335	534	466	359	469
*GOF*	1.028	1.021	0.968	1.011	1.007
*R*_1_*^a^* [*I*_o_>2σ*I*)]	0.0471	0.0645	0.0918	0.0716	0.0562
*wR*_2_*^b^* [*I*_o_>2σ*I*)]	0.1169	0.1526	0.1782	0.1948	0.0991

*^a^ R*_1_ = Σ||*Fo*| − |*Fc*||/Σ|*Fo*|. *^b^ wR*_2_ = {Σ[*w*(*Fo*^2^ − *Fc*^2^)^2^]/Σ[*w*(*Fo*^2^)^2^]}^1/2^.

## Data Availability

The data presented in this study are available on request from the corresponding author.

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
