# Peer review of "Third Generation Buchwald Precatalysts with XPhos and RuPhos: Multigram Scale Synthesis, Solvent-Dependent Isomerization of XPhos Pd G3 and Quality Control by 1H- and 31P-NMR Spectroscopy"

_molecules, 2021, doi:10.3390/molecules26123507_

Round 1

Reviewer 1 Report

This manuscript by Kolotilov and co-workers describes a method development for estimating the purity of the Buchwald precatalysts. Buchwald precatalysts Pd(ABP)(Phos)(OMs) (with XPhos and RuPhos were prepared and were analyzed using X-ray crystallography and NMR spectroscopy. The feasibility of 1H and 31P NMR spectroscopy to determine the percentage of impurities in these compounds was demonstrated in this work. In this work, the authors discovered that Phos Pd G3 precatalysts can undergo isomerization in solution and exist in two forms. Overall, this manuscript is well written. The data are discussed in details. Given the importance of Buchwald precatalysts in the field, this work provides useful information. The reviewer recommends publication of this work in Molecules with minor revisions listed below:

  1. In the abstract, change “was performed” to “were prepared”
  2. Page 2, Figure 1, the “PCy2” substituents can be drawn more appropriately with the right bond angle (120 o).
  3. Page 4, line 152, remove “this” from “the this”.
  4. Charges on ions should be written as, for example, Pd(II) or Pd2+, but nor PdII
  5. Page 19, line 524, “CHCl3 or CH2Cl2” should be CDCl3 or CD2Cl2

Author Response

Reviewer 1.

COMMENT

  1. In the abstract, change “was performed” to “were prepared”

REPLY

We meant that "Multi-gram scale synthesis… was performed". From the Reviewer's comment we see that the sentence was confusing, thus we re-phrased the whole sentence in order to make it more clear and unambiguous.

COMMENT

  1. Page 2, Figure 1, the “PCy2” substituents can be drawn more appropriately with the right bond angle (120 o).

REPLY

We changed the Figure as recommended by the Referee.

COMMENT

  1. Page 4, line 152, remove “this” from “the this”.

REPLY

We changed this, we thank the Referee for pointing out our mistake.

COMMENT

  1. Charges on ions should be written as, for example, Pd(II) or Pd2+, but nor PdII

REPLY

Following recommendation of the Referee, we changed "PdII" into "Pd2+" in those cases when this symbol referred to single ion with formal oxidation state 2+. At the same time we changed "PdII" into "palladium(II)" in the cases when this symbol referred to the oxidation state of this element in the complex.  

COMMENT

  1. Page 19, line 524, “CHCl3 or CH2Cl2” should be CDCl3 or CD2Cl2

REPLY

The solutions were used not to 1H, but for 31P spectra measurements. We used non-deuterated solvents.

Reviewer 2 Report

In the present study authors report on the investigation of Buchwald precatalysts Pd(ABP)(Phos)(OMs) with 2-dicyclohexylphosphino-2′,4′,6′-triisopropylbiphenyl (XPhos) and 2-dicyclohexylphosphino-2′,6′-diisopropoxybiphenyl (RuPhos) by 1H and 31P NMR technique. It was demonstrated the possibility of estimating the purity of precatalysts on the basis of their NMR spectra. Reversible isomerization of Pd(ABP)(Phos)(OMs) upon solvent change was also studied.  Also authors have isolated and characterized several impurities in the precatalysts. The manuscript is well written, material is of interest for a wide scientific Audience of Journal Molecules, and itmerits the publication after some small revisions:

Small corrections:

  • Figures 11 and 12. Please change cyrillic symbols м.д. with ppm.
  • Figures S3-S17 of ESI. The same: м.д. --> ppm 
  • Please, add the NMR spectral data for compounds 3 and 4 to the experimental part (chapter 4.2.2).

Author Response

COMMENT

Figures 11 and 12. Please change cyrillic symbols м.д. with ppm.

REPLY

We changed cyrillic letters by "ppm". We thank to the Referee for pointing out our mistake.

COMMENT

Figures S3-S17 of ESI. The same: м.д. --> ppm 

REPLY

We changed cyrillic letters by "ppm". We thank to the Referee for pointing out our mistake.

COMMENT

Please, add the NMR spectral data for compounds 3 and 4 to the experimental part (chapter 4.2.2).

REPLY

We added the data as recommended by the Referee.